# The Mitochondrial Genome of the Globally Invasive Barnacle *Megabalanus coccopoma* Darwin 1854 (Crustacea: Balanomorpha): Rearrangement and Phylogenetic Consideration within Balanomorpha



Mengjuan Zhang [1,2], Yuefeng Cai [1,2,3,*], Nanjing Ji [1,2,3], Benny Kwok Kan Chan [4] and Xin Shen [1,2,3,*]

1   Jiangsu Key Laboratory of Marine Bioresources and Environment, Jiangsu Ocean University, Lianyungang 222005, China
2   Jiangsu Key Laboratory of Marine Biotechnology, Jiangsu Ocean University, Lianyungang 222005, China
3   Co-Innovation Center of Jiangsu Marine Bio-Industry Technology, Jiangsu Ocean University, Lianyungang 222005, China
4   Biodiversity Research Center, Academia Sinica, Taipei 115, Taiwan
*   Correspondence: yuefengcai@jou.edu.cn (Y.C.); shenthin@163.com (X.S.); Tel.: +86-518-85895121 (X.S.)

**Abstract:** *Megabalanus coccopoma* (Darwin, 1854) is a globally invasive species in Balanomorpha (Crustacea). This species is a model organism for studying marine pollution and ecology. However, its mitogenome remains unknown. The mitogenome sequencing of *M. coccopoma* is completed in the present study. It has a 15,098 bp in length, including 13 protein-coding genes (PCGs), 2 ribosomal RNAs (rRNAs), 22 transfer RNAs (tRNAs), along with a putative regulatory area. A substantial A+T bias was observed in the genome composition (68.2%), along with a negative AT (0.82) and GC ($-0.136$) skew. Compared to the gene sequence of the ground model of pan-crustacea, 13 gene clusters (or genes), such as 10 tRNAs and 3 PCGs, were observed in a different order. This was in line with the previously observed large-scale gene rearrangements of Balanomorpha. Among the 37 genes, the gene cluster (*M-nad2-W-cox1-$L_2$-cox2-D-atp8-atp6-cox3-G- nad3-R-N-A-E-$S_1$*) Balanomorpha was conserved. Furthermore, phylogeny analysis indicated that the existing Balanomorpha species family was divided into nine rearrangement patterns, supporting the polyphyly of Balanoidea.

**Keywords:** *Megabalanus coccopoma*; mitochondrion; gene rearrangement; phylogeny

## 1. Introduction

Barnacles are important model organisms in marine ecology and biofouling studies in the superorder Thoracicalcarea (Crustacea: Thecostraca) [1]. The Balanomorpha order is a highly evolved, complex, and morphologically differentiated Thoracicalcarea barnacle [1,2]. However, the phylogeny of this group, particularly at higher taxonomic levels, is poorly understood [3]. *Megabalanus coccopoma* (Darwin, 1854) belongs to the Balanidae of Thoracicalcarea and is an invasive global species [4]. *M. coccopoma* is native to the tropical eastern Pacific [5]. Several studies on *M. coccopoma* have focused on larval development and species distribution [6]. However, its molecular characteristics are relatively unknown. *M. coccopoma* presents considerable difficulties in accurate species identification and phylogenetic relationship determination. This is because of the significant variation in the shell morphology during development and response to the habitat. The taxa of balanomorphan species have been controversial. Using complete mitogenomes in species identification is increasingly common, particularly for contested taxa with a similar outward appearance [7]. However, no studies have been published describing the entire mitochondrial genome of *M. coccopoma*.

The mitochondrial genome is a reliable and effective molecular marker in biological phylogenetic studies [8]. The mitochondrial genome sequence can reveal Balanomorph

evolution. There are 37 genes within the circular mitochondrial genome of balanomorphan barnacles, with 13 protein-coding genes (PCGs), 22 transport RNAs, and 2 ribosomal RNAs. Some Balanomorph species could have more or less than 37 mitochondrial genes. For instance, a unique $S_2$-C-Y repeat was identified in *Epopella plicata* of the Tetraclitidae [9]. Moreover, a *C* deletion was found in *Chthamalus malayensis* in the Chthamalidae [10]. In a few cases, the number of protein-coding genes may also vary other than changes in the number of genes transporting RNA [11].

The arrangement of mitochondrial genes could effectively analyze the phylogenetic relationship between species [12]. The Balanomorph mitochondrial genome underwent significant gene rearrangements involving smaller transport RNA and gene blocks with multiple PCGs and transport RNA [13]. Lim and Hwang (2006) sequenced the entire mitochondrial genome of the pollicipedid *Capitulum mitella*. They compared it with *Pollicipes polymerus*, *Tetraclita japonica,* and *Megabalanus volcano*. Moreover, they found that there is a Thoracicacalcarea-specific translocation, particularly (1) *K-Q* between the control region and *I*, (2) *A-E* between *N* and $S_1$, (3) *P* between *nd4l* and *T,* and (4) *Y-C* between $S_2$ and *nd1* [14]. Subsequently, the first mitochondrial whole genome was determined from the coral symbiotic pyrgomatid, *Nobia grandis*. At least seven transporter RNAs rearranged in mitochondrial genes were compared to the plesiomorphic gene arrangement order of pancrustaceans [15]. Shen et al. performed mitochondrial whole-genome sequencing of *Epopella plicata*. They identified the gene string $S_2$-C-Y unique repeats and raised questions concerning the Balanoidea monophyly [9]. In the same year, Shen et al. further sequenced the whole mitochondrial genome of *Tetraclita serrata*. Based on gene alignments and phylogenetic analyses, they demonstrated the non-monophyletic nature of Balanidae. However, it is necessary to establish the non-monophyly of the barnacle family due to the low bootstrap values in the evolutionary tree [16]. This is supported by the growing evidence pointing to an ancestral arrangement of mitochondrial genes in the Balanomorpha. However, the mitochondrial gene plesiomorphic arrangement pattern in Balanomorpha is unknown [17]. Therefore, the Balanomorpha mitochondrial genome rearrangement study can provide vital information on the profound systemic evolution within the Balanomorpha collective group [18–20]. However, the Balanomorpha mitochondrial genome data are still scarce, seriously restricting understanding of the systematic evolution across various groups [21,22].

In this study, (i) the entire mitogenome of an *M. coccopoma* individual was sequenced to provide a comprehensive description of the findings, (ii) the gene rearrangement in the mitogenome of Balanomorpha was addressed, and (iii) a Thoracicalcarea phylogenetic tree was obtained to determine the relationship of *M. coccopoma* with other barnacles.

## 2. Materials and Methods

### 2.1. Sample Collection and DNA Extraction

The *M. coccopoma* specimen was obtained from Haizhou Bay, situated in Ganyu District, Lianyungang City, Jiangsu Province, China (119.208767° N, 34.944531° E). It was kept in the Museum of Ocean University, Jiangsu, China (voucher number: TkuLYG-012). The whole genomic DNA was extracted from 95% ethanol-preserved muscle tissue with the TIANamp Marine Animal DNA Kit (TIANGEN, China) based on the manufacturer's instructions. The collected specimen was classified based on the morphological characteristics provided by Liu and Ren (2007), other than partially sequencing the *cox1* and *H3* genes [23].

### 2.2. PCR and Sequence Determination

Universal and specific primers were employed to amplify the *cox1* and Histone 3 (*H3*) genomic regions (Table 1). The reaction mixture for amplifying the *cox1* and H3 gene fragments comprised 1 μL of DNA template, 12.5 μL of ExTaq polymerase, 9.5 μL of sterile distilled water, and 1 μL of each primer. The *cox1* gene amplification was carried out with the following cycle parameters: initial denaturation at 94 °C for 5 min, followed by denaturation for 35 cycles at 94 °C for 1 min; annealing at 40 °C for 1 min; further

elongation at 72 °C for 1.5 min; and, ultimately, extension at 72 °C for 7 min. The H3 gene was denatured for 3 minutes at 95 °C. Then, we performed 35 cycles of denaturation for 30 s at 95 °C, annealing at 58 °C, elongation at 72 °C for 1 minute, and extension for 5 minutes at 72 °C. The PCR results obtained from the *cox1* and H3 genes were sent to Shanghai Maple Biological Company for further sequence analysis.

**Table 1.** Mitochondrial gene amplification employs general and highly specialized primers.

| Primer Name | Sequence (5′–3′) |
| --- | --- |
| dgLCO | GGTCAACAAATCATAAAGAYATYGG |
| dgHCO | TAAACTTCAGGGTGACCAAARAAYCA |
| H3-F1 | ATGGCTCGTACCAAGCAGACVGC |
| H3-R1 | ATATCCTTRGGCATRATRGTGAC |

Note: Degenerate bases: R = A/G, Y = C/T, V = C/A/G.

The DNA was ultrasonically fragmented into 350 bp fragments after analyzing and quantifying the DNA samples. After this step, the DNA fragments were subjected to end repair by adding a 3′-end A nucleotide and a sequencing junction. The fragments were purified, and PCR amplification was performed to build the sequencing library. After going through Illumina's quality control, the libraries were sequenced by Illumina HiSeq 4000 instruments. The following served as an outline for the step-by-step experimental procedure: data processing involves checking the quality (removing connectors and data of low quality), comparing the data to a reference genome, and assembling the mitochondrial genome, followed by its annotation.

### 2.3. Gene Identification

Sequence splicing was performed with Geneious [24] and rechecked using SeqMan [25] for the whole mitochondrial genome length of the species. The preliminary gene prediction was made using the MITOS WebServer [26]. The tRNAscan-SE 1.21 [27] online website for preliminary gene predictions of mitochondrial genes was individually selected and blast-compared using the NCBI website. The gene annotation was finalized by combining the corresponding characteristics of each gene with other Cirripedia species from the database. The annotated sequences were submitted online through the BankIt platform in NCBI. The full length, base composition, base offset, gene number, coding strand, codon usage, amino acid usage, non-coding regions, and control regions of mitochondrial DNA were obtained from the PhyloSuite software [28]. Mitochondrial genomes were applied through the OGDRAW [29] online website.

### 2.4. Genome Analysis and Phylogenetic Analysis

According to the pancrustacean plesiomorphic arrangement, the mitochondrial gene order of 34 species from the Balanomorpha and *Altiverruca navicular* in Verrucomorpha (Supplementary Table S1) was mapped. The gene order of these 35 species was separated into 10 models based on the original gene order (Model 1–10). The gene rearrangements were analyzed using CREx [30].

The 44 mitochondrial genomes from Cirripedia and the newly obtained mitochondrial genome sequence of *M. coccopoma* were used for phylogenetic analysis (Supplementary Table S2). The species sequences downloaded from NCBI were imported into PhyloSuite, and multiple sequence alignment was performed with MAFFT [31]. The tree model of the IQ-TREE [32] was determined using ModelFinder [33]. The protein-coding genes of the barnacle mitochondrial genome helped construct the IQ-TREE (main parameters: Model, GTR; Bootstrap, 1000; and Outgroup, *Polyascus gregaria*). The editing and beautification of the phylogenetic tree were completed on Interactive Tree Of Life (iTOL) [34–36].

## 3. Results

### 3.1. General Characteristics

The ring-shaped mitochondrial genome of *M. coccopoma* is a 15,098 bp molecule, the same as other Balanomorpha barnacles. Besides 13 PCGs genes, 22 tRNA genes and 2 rRNA genes exist. The complete details are reported in (Table 2, Figure 1). The heavy chain (H chain) of *M. coccopoma* encodes 30 genes, while the light chain (L chain) encodes 7. The light chain encodes the PCG *nd1* and 2 ribosomal RNAs, while the heavy chain encodes the other 12 PCGs. There are seven gene overlaps within the mitochondrial genome of *M. coccopoma*. Seven bases overlap between *atp8* and *atp6*, *nd4* and *nd4L*, and one or two in the other genes. The mitochondrial genome of *M. coccopoma* has a non-coding region of 546 bp, the longest of which is located between *srRNA* and *I* and has 415 bp. Additionally, it is located in the regulatory area within the mitochondrial genomes of other genera of Balanomorpha. The mitochondrial genome sequence of *M. coccopoma* was submitted to GenBank with the accession number OK631889.

**Table 2.** The mitochondrial genome of *M. coccopoma*.

| Gene | Stand | Position | | Nucleotide | Codons | | Anti-Codon | Intergenic Sequence * |
| | | Start | Stop | | Start | Stop | | |
|---|---|---|---|---|---|---|---|---|
| *cox1* | H | 10 | 1545 | 1,536 | CGA | TAA | | 2 |
| $L_2$ | H | 1548 | 1615 | 68 | | | taa | 5 |
| *cox2* | H | 1621 | 2304 | 684 | ATG | TAA | | 0 |
| *D* | H | 2305 | 2368 | 64 | | | gtc | 0 |
| *atp8* | H | 2369 | 2527 | 159 | ATT | TAA | | −7 |
| *atp6* | H | 2521 | 3186 | 666 | ATG | TAA | | −1 |
| *cox3* | H | 3186 | 3972 | 787 | ATG | T- | | 0 |
| *G* | H | 3973 | 4036 | 64 | | | tcc | 0 |
| *nad3* | H | 4037 | 4388 | 352 | ATT | T- | | 0 |
| *R* | H | 4389 | 4451 | 63 | | | tcg | 0 |
| *N* | H | 4452 | 4515 | 64 | | | gtt | 0 |
| *A* | H | 4516 | 4581 | 66 | | | tgc | 1 |
| *E* | H | 4583 | 4648 | 66 | | | ttc | 0 |
| $S_1$ | H | 4649 | 4707 | 59 | | | gct | 75 |
| *P* | H | 4783 | 4846 | 64 | | | tgg | 0 |
| *nad4L* | H | 4847 | 5140 | 294 | GTG | TAA | | −7 |
| *nad4* | H | 5134 | 6463 | 1330 | ATG | T- | | 0 |
| *H* | H | 6464 | 6528 | 65 | | | gtg | 0 |
| *nad5* | H | 6529 | 8230 | 1702 | ATT | T- | | 0 |
| *F* | H | 8231 | 8294 | 64 | | | gaa | 5 |
| *T* | H | 8300 | 8366 | 67 | | | tgt | 0 |
| *nad6* | H | 8367 | 8855 | 489 | ATG | TAA | | −1 |
| *cob* | H | 8855 | 9994 | 1140 | ATG | TA- | | −2 |
| $S_2$ | H | 9993 | 10,062 | 70 | | | tga | 24 |
| *Y* | H | 10,087 | 10,150 | 64 | | | gta | 24 |
| *K* | L | 10,175 | 10,240 | 66 | | | ttt | 9 |
| *Q* | L | 10,250 | 10,317 | 68 | | | ttg | 1 |
| *C* | L | 10,319 | 10,380 | 62 | | | gca | −2 |
| *nad1* | L | 10,379 | 11,305 | 927 | ATA | TAA | | −3 |
| $L_1$ | L | 11,303 | 11,370 | 68 | | | tag | 0 |
| *rrnL* | L | 11,371 | 12,689 | 1319 | | | | 0 |
| *V* | L | 12,690 | 12,737 | 48 | | | tac | 1 |
| *rrnS* | L | 12,739 | 13,486 | 748 | | | | 415 |
| *I* | H | 13,902 | 13,969 | 68 | | | gat | 0 |
| *M* | H | 13,970 | 14,035 | 66 | | | cat | 0 |
| *nad2* | H | 14,036 | 15,034 | 999 | ATG | TAA | | −2 |
| *W* | H | 15,033 | 15,098 | 66 | | | tca | 9 |

* Negative numbers depict the overlapping nucleotides between the adjacent genes.

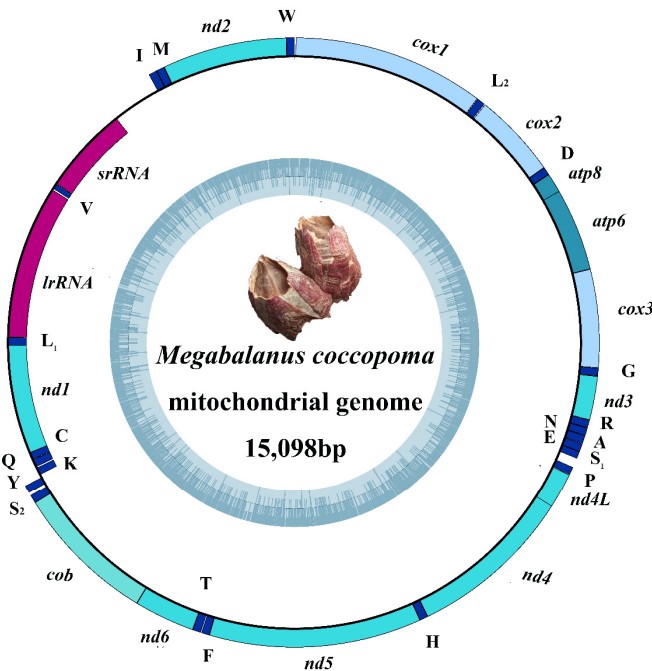

**Figure 1.** The genetic map of the *M. coccopoma* mitochondrial genome. The GC content is displayed within the inner ring. Genes encoded on the exterior appeared on heavy strands, while the interior-encoded genes appeared on light strands.

### 3.2. Codon Use and The Genetics of Protein Synthesis

The 13 PCGs observed in *M. coccopoma* have a nucleotide sequence of 11,056 bp long. It accounts for 73.22 % of the total mitochondrial genome length. All 13 PCGs in *M. coccopoma* begin with ATN (N=A, T, G, and C), except for *nd4L*, which has GTG as the start codon. This reflects the conservation in Balanomorpha genes. In addition, five PCGs (*cox3*, *nd3*, *nd5*, *nd4*, and cob) terminate with incomplete terminators (TA- or T-). In contrast, the remaining PCGs are terminated using complete TAA termination. (Table 2). Moreover, incomplete terminators are prevalent in Balanomorpha [15,37,38]. The 13 PCGs of *M. coccopoma* have 3,663 codons (without the incomplete terminators), with the six most frequent ones being UUU > AUU > UUA > AUA > UCU > UUC, 2.94% to 7.05%, and the lowest being CGC (0.11%). The multiple codons are UUU (259), followed by AUU (257) (Figure 2, Supplementary Table S3). The most commonly utilized amino acids were leucine (Leu) (544), phenylalanine (Phe) (367), serine (Ser) (359), and isoleucine (Ile) (317).

### 3.3. Composition of The Base and Skew

The relative quantity of nucleotides A and T can identify the variation in base composition found in each gene, indicated as the AT and GC skew [39]. In the mitochondrial genome sequence of *M. coccopoma*, nucleotide A involves 31.3% of the total. In comparison, nucleotide C makes up 18.1%, nucleotide G makes up 13.8%, and nucleotide T makes up 36.9%. *Atp8* genes had the highest proportion of A codons than the other 13 PCGs (34.6%). The *cox3* and *cob* genes had the highest C content at 20.8%, *nd1* had the highest G content at 15.8%, and *nd1* possessed the highest T content at 46.0%. The A + T content ranged between 64.5% (*cox2*) and 71.7% (*atp8*). All 13 PCGs revealed T to A shifts ranging from 0.035 (*atp8*) to 0.292 (*nd1*). Two protein-coding genes had G to C (*nd1* and *nd4L*) ranging from 0.004 (*nd4L*) to 0.236 (*nd1*), and another eleven PCGs possessed C to G from 0.01 (*nd4*) to 0.478 (*nd6*) (Table 3). Previous studies demonstrated that the AT content of Metazoa species is usually higher than the GC level [40].

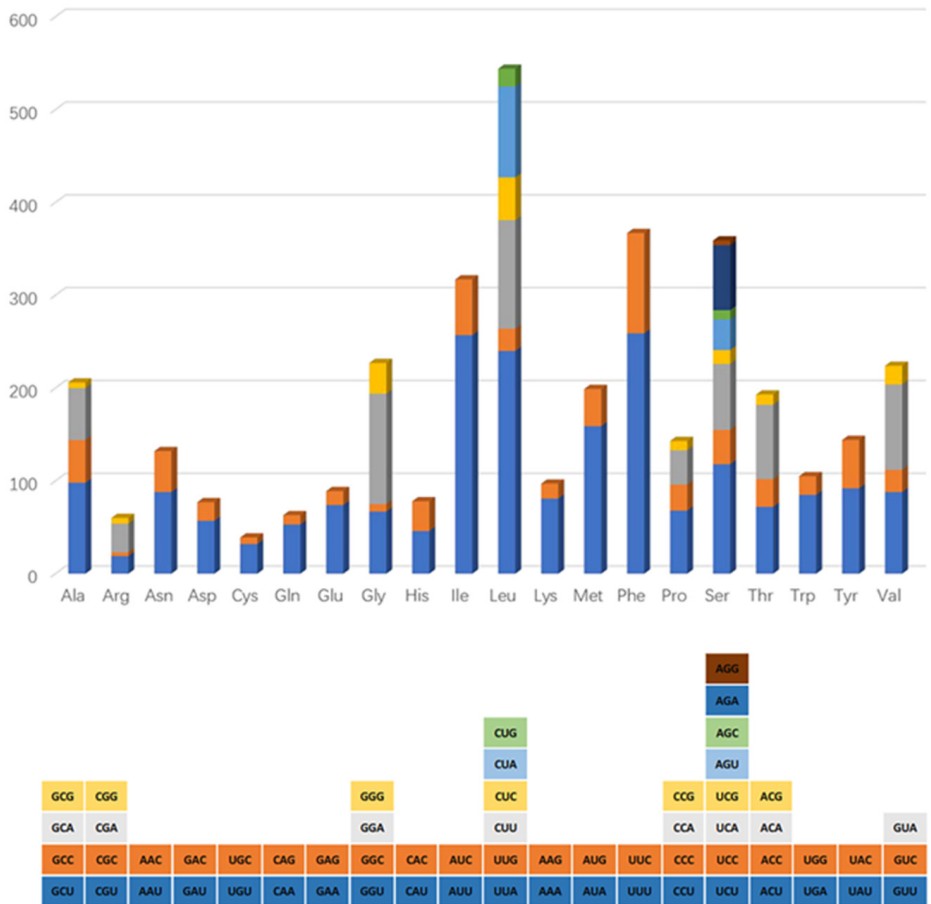

**Figure 2.** Codon usage among the 13 mitochondrial protein-coding genes of *M. coccopoma*.

**Table 3.** Nucleotide composition and skew of *M. coccopoma* mitochondrial genes.

| Gene | Nucleotide Proportion | | | | A + T (%) | AT Skew | GC Skew |
|---|---|---|---|---|---|---|---|
| | A (%) | C (%) | G (%) | T (%) | | | |
| atp6 | 30.3 | 18.6 | 11.0 | 40.1 | 70.4 | −0.139 | −0.257 |
| atp8 | 34.6 | 20.1 | 8.2 | 37.1 | 71.7 | −0.035 | −0.420 |
| cob | 27.0 | 20.8 | 14.5 | 37.7 | 64.7 | −0.165 | −0.178 |
| cox1 | 27.5 | 18.6 | 16.9 | 36.9 | 64.5 | −0.146 | −0.048 |
| cox2 | 30.6 | 19.7 | 14.0 | 35.7 | 66.2 | −0.077 | −0.169 |
| cox3 | 26.3 | 20.8 | 15.8 | 37.1 | 63.4 | −0.170 | −0.137 |
| nd1 | 25.2 | 11.0 | 17.8 | 46.0 | 71.2 | −0.292 | 0.236 |
| nd2 | 28.2 | 17.8 | 13.7 | 40.2 | 68.5 | −0.175 | −0.130 |
| nd3 | 28.7 | 18.2 | 12.5 | 40.6 | 69.3 | −0.172 | −0.186 |
| nd4 | 27.4 | 15.6 | 15.3 | 41.6 | 69.0 | −0.206 | −0.010 |
| nd4L | 29.4 | 14.3 | 15.5 | 40.8 | 70.2 | −0.162 | 0.040 |
| nd5 | 28.6 | 16.9 | 15.7 | 38.8 | 67.5 | −0.151 | −0.037 |
| nd6 | 31.7 | 22.1 | 7.8 | 38.5 | 70.1 | −0.097 | −0.478 |
| srRNA | 34.4 | 12.8 | 21.5 | 31.3 | 65.6 | 0.047 | 0.254 |
| lrRNA | 36.4 | 10.2 | 16.9 | 36.6 | 73.0 | −0.003 | 0.247 |
| All PCGs | 28.2 | 17.8 | 14.8 | 39.3 | 67.5 | −0.164 | −0.092 |
| All genes | 31.3 | 18.1 | 13.8 | 36.9 | 68.2 | −0.082 | −0.136 |

### 3.4. Genes for Ribosomal and Transfer RNA

The 22 transport RNAs encoded by the mitochondrial genomes of *M. coccopoma* fold into cloverleaf secondary structures of varying sizes. This is consistent with the anticodon utilization of most known Balanomorpha species. The *M. coccopoma srRNA* gene is positioned between the non-coding region and the *V* gene, which is 748 bp long, and the *lrRNA* gene is situated between the $L_1$ and *V* genes, which are 1319 bp long. Light-strand encoding of *lrRNA* and *srRNA* leads to 73.0 and 65.6% A+T, respectively. The AT skew and GC offset of the *lrRNA* gene were, respectively, $-0.003$ and 0.247. In contrast, the AT and GC skews of the *srRNA* gene were $-0.164$ and 0.254, depicting T-A and G-C skew (Table 3).

### 3.5. Gene Arrangement

Based on the plesiomorphic order of pancrustaceans, the present study mapped and classified the mitochondrial gene order of 35 barnacles (Figure 3). This yielded 10 different types, each with a unique conserved gene block. Compared to the original crustacean arrangement, *M. coccopoma* revealed translocations in four genes (*A*, $S_1$, *T*, *Q*, and *C*), inversions in two (*Y* and *K*), and seven conserved gene blocks (*cox1-$L_2$-cox2, D-atp8-atp6-cox3-G-nad3, R-N, F-nad5-H-nad4-nad4L, nad6-cob-$S_2$, nad1-$L_1$-lrRNA-V-srRNA-I,* and *M-nad2-W*). Furthermore, *M. coccopoma* indicates an inversion (from light to heavy strand) of the gene cluster (*P-nd4L-nd4-H-nd5-F*) with three PCGs and three transporter RNAs. This gene rearrangement is consistent with previous studies [41–43]. The Balanidae, in which *M. coccopoma* is found, has three Models: Model 1, Model 3, and Model 6. *M. coccopoma* shares Model 3 with Balanidae (*M. volcano*, *Megabalanus tintinnabulum*, *Megabalanus ajax*, *Acasta cyathus*, *Acasta sulcata*, and *Balanus trigonus*). The Pyrgomatidae is Model 1. In this gene arrangement, the Tetraclitidae are predominantly Model 2, Austrobalanidae (Model 7), Chelonibiidae (Model 8), Catophragmidae (Model 4), Chthamalidae (Model 4, Model 5, and Model 9), Chionelasmatidae (Model 4), and Verrucidae (Model 10). In addition, the 10-type gene arrangement order is only conserved in the genera and not between different families. For instance, the Balanidae (*Armatobalanus allium*, *Striatobalanus amaryllis*, *Amphibalanus amphitrite*, *Balanus Balanus*, *Fistulobalanus albicostatus*, and *Semibalanus cariosus*) and the Pyrgomatidae (*Nobia grandis*, *Pyrgopsella youngi*, and *Savignium* sp. BKKC-2014) depict the same gene arrangement model (Figure 3, Model 1). The Chthamalidae (*Octomeris* sp. BKKC-2014), the Catophragmidae (*Catomerus polymerus*), and the Chionelasmatidae (*Eochionelasmus ohtai* and *Eochionelasmus coreana*) have the same gene arrangement model (Model 4). In the phylogenetic tree, the different gene arrangements are well clustered. For example, *Balanus trigonus* (Model 3) and *A. amphitrite* (Model 1) cluster together, and *Semibalanus cariosus* (Model 1) and *Striatobalanus amaryllis* (Model 1) cluster together. Therefore, the order of Balanomorpha genes is consistent with the results of the phylogenetic tree (Figure 4). The nine model gene arrangements in the barnacle suborder indicate that two gene clusters (*cox1-$L_2$-cox2-D-atp8-atp6-cox3-G-nad3-R-N-A-E-$S_1$*) and the gene cluster (*M-nad2-W*) are conserved in the barnacle suborder through a consistent structure. Additionally, these two gene clusters could be combined to form a single lengthy gene cluster (*M-nad2-W-cox1-$L_2$-cox2-D-atp8-atp6-cox3-G-nad3-R-N-A-E-$S_1$*) since the mitochondrial genome of the Balanomorpha is circular. It can be observed that this specific connection may exist between different models, requiring more mitochondrial gene arrangement order of Balanomorpha species. Therefore, it is possible to infer a primitive arrangement in the barnacle suborder, and this connection may exist between different models.

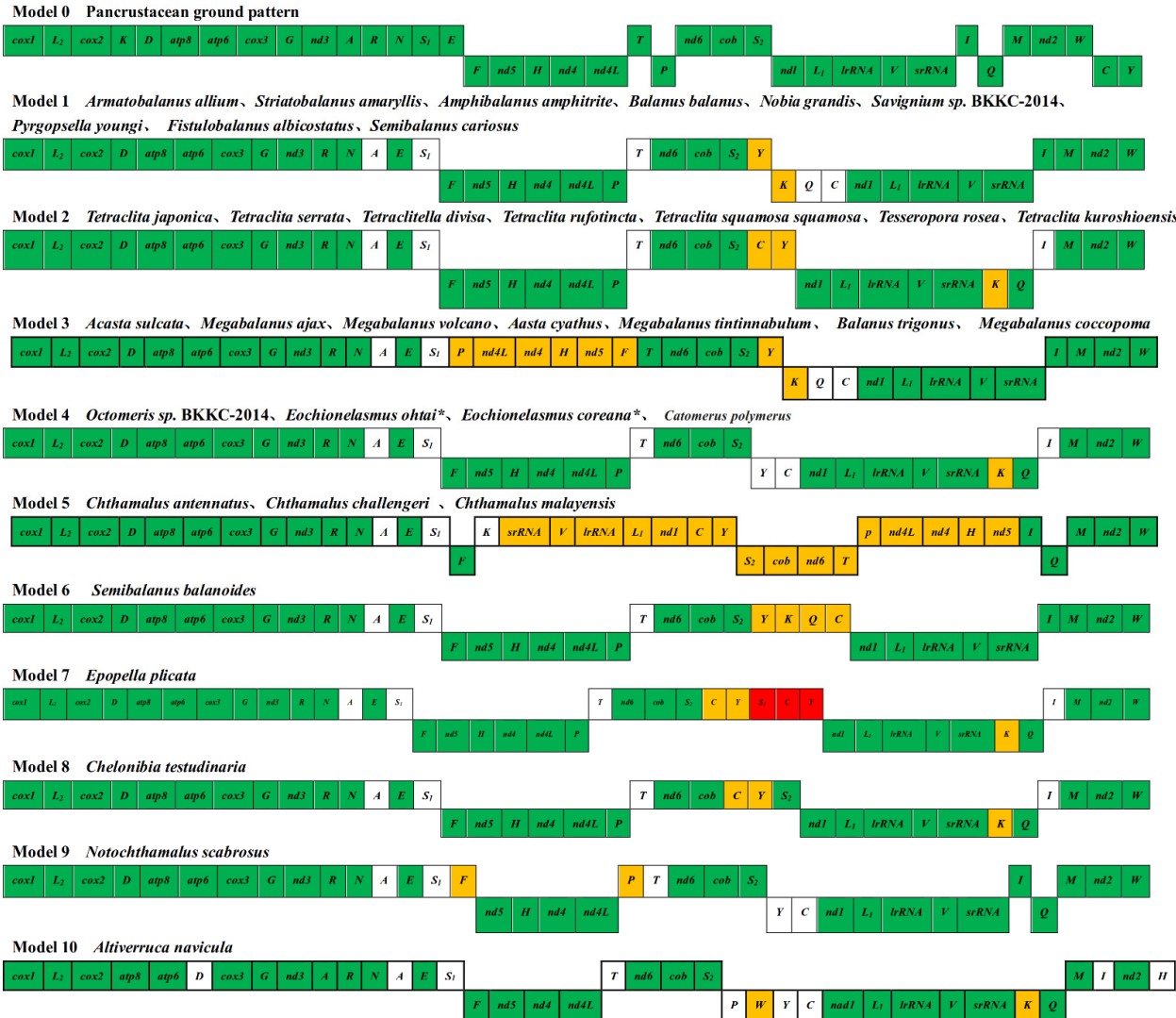

**Figure 3.** Comparison of mitochondrial gene arrangements in the Balanomorpha suborder. Shadow boxes represent conserved blocks; orange and green indicate blocks with or without inversion, respectively; white depicts translocated blocks; and encoded in heavy and light chains, genes are placed on the top and bottom line, respectively.

While examining the gene arrangement order, we found it highly differentiated in Balanomorpha. A comparison with the original gene arrangement order revealed conserved blocks between different families due to the specificity of the mitochondrial gene ring structure. Still, the evolutionary tree results did not cluster the taxa depending on these blocks. The "duplication/random loss" mechanism is currently the most well-recognized theory and is frequently utilized to explain translocations in the mitochondrial genome. However, this mechanism causes duplication of gene areas, followed by deletion of some of the tandem duplicated regions due to mismatches in the downstream strand during replication [44]. Consequently, the duplication–random loss and recombination models explain the observed large-scale gene rearrangements. In many cases, repeated random loss (TDRL) events could indicate the evolutionary direction of rearrangements. This would allow the ancestral state reconstruction by comparing two gene orders without accounting for the outgroups. Using the CREx estimated scenario, we investigated the mitochondrial gene rearrangements, focusing on the TDRL event, across 34 species belonging to the Balanomorpha and 1 species belonging to the Verrucomorpha. Balanomorpha can be

divided into three TDRL events, TDRL(a), TDRL(b), and TDRL(c), while Verrucomorpha is
TDRL(d) (Figure 5).

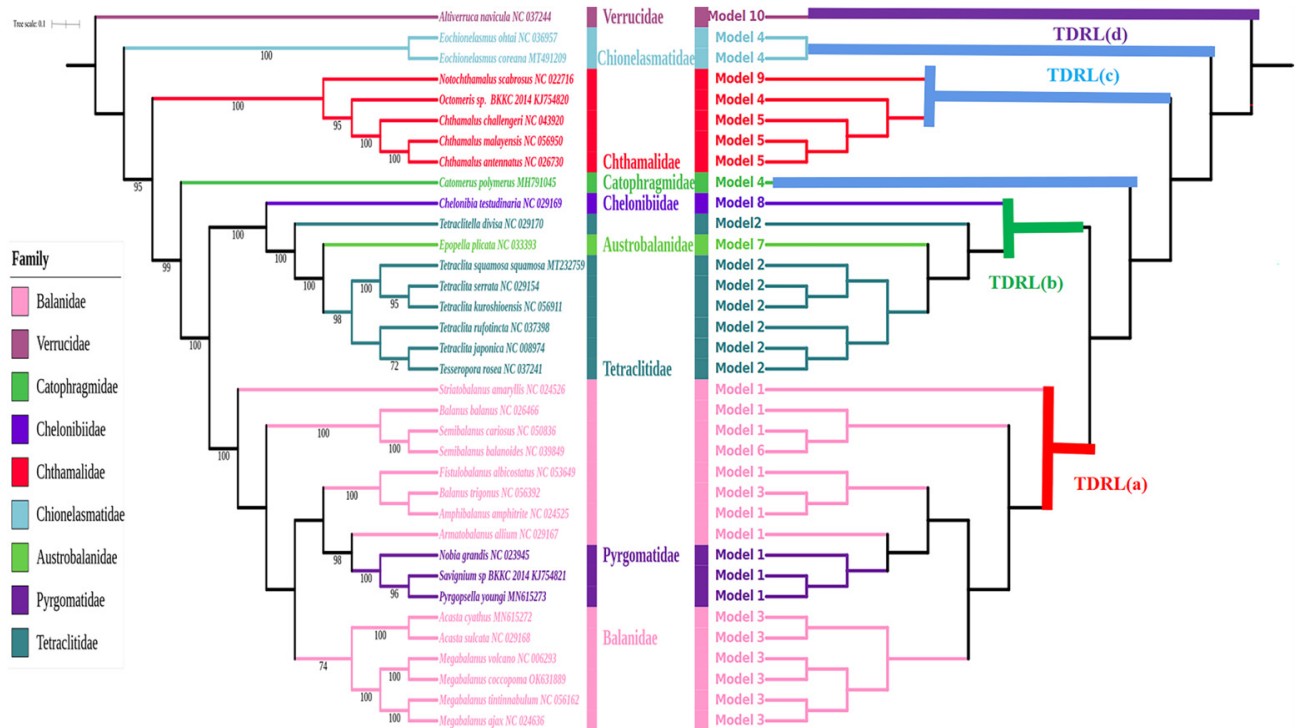

**Figure 4.** The IQ-Tree is based on 13 PCGs nucleotide sequences of *M. coccopoma* and other mitochondrial genomes from 35 barnacles. The numbers at the nodes represent the bootstrap values derived from ML analysis. The IQ-TREE will disregard inverted branch lengths, and TDRL events will be omitted.

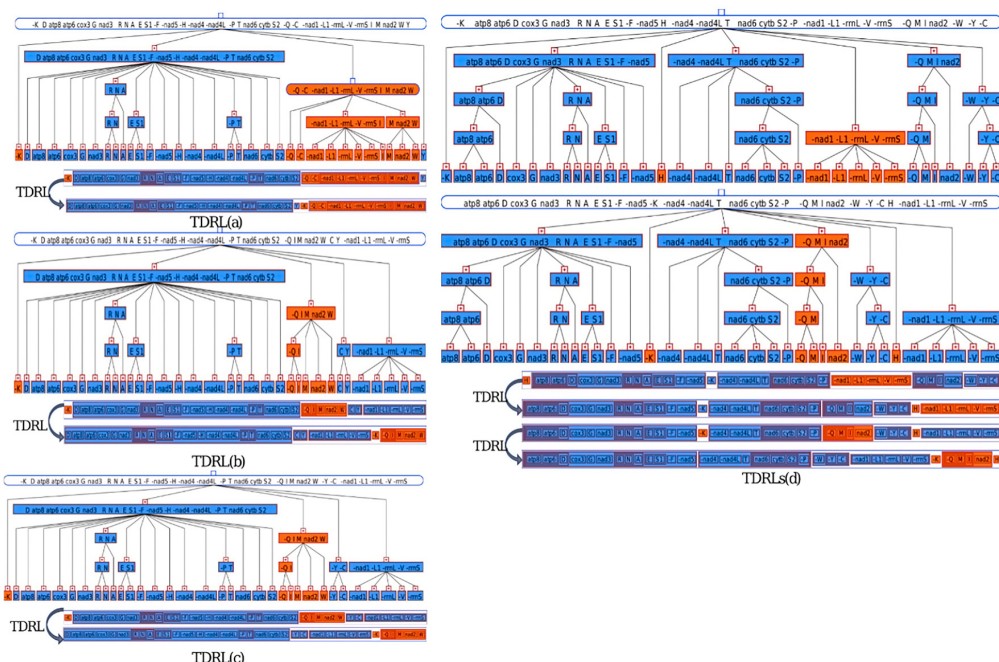

**Figure 5.** CREx estimated the scenario of mitochondrial gene order rearrangement (TDRLs) for the given phylogeny of the Balanomorpha order.

Model 1 starts with the transposition of genes (*A*, *S₁*, and *T*), followed by TDRL events of gene clusters (*nad1-L₁-lrRNA-V-srRNA-I-Q-M-nad2-W-C*) which is different from the plesiomorphic pancrustacean arrangement. Consecutive copies of the gene cluster *nad1-L₁-lrRNA-V-srRNA-I-Q-M-nad2-W-C* are followed by randomly losing duplicate genes. The first consecutive copies of the *nad1-L₁-lrRNA-V-srRNA-I-Q-M-nad2-W-C-nad1-L₁-lrRNA-V-srRNA-I-Q-M-nad2-W* gene cluster are followed by the accidental loss of gene blocks *nad1-L₁-lrRNA-V-srRNA-I*, *M-nad2-W*, *Q* and *C* to form gene cluster *Q-C-nad1-L₁-lrRNA-V-srRNA-I-M-nad2-W*. This is followed by gene (*Y* and *K*) inversions and, finally, by TDRL events of gene clusters (*K-D-atp8-atp6-cox3-G-nad3-R-N-A-E-S₁-F-nad5-H-nad4-nad4L-P-T-nad6-cob-S₂-Q-C-nad1-L₁-lrRNA-V-srRNA-I-M-nad2-W-Y*). First, the consecutive copies are *K-D-atp8-atp6-cox3-G-nad3-R-N-A-E-S₁-F-nad5-H-nad4-nad4L-P-T-nad6-cob-S₂-Q-C-nad1-L₁-lrRNA-V-srRNA-I-M-nad2-W-Y-K-D-atp8-atp6-cox3-G-nad3-R-N-A-E-S₁-F-nad5-H-nad4-nad4L-P-T-nad6-cob-S₂-Q-C-nad1-L₁-lrRNA-V-srRNA-I-M-nad2-W-Y*. This is followed by the random loss of gene blocks, such as *K*, *Q-C-nad1-L₁-lrRNA-V-srRNA-I-M-nad2-W*, *D-atp8-atp6-cox3-G-nad3-R-N-A-E-S₁-F-nad5-H-nad4-nad4L-P-T-nad6-cob-S₂,* and *Y*, forming a gene cluster (*D-atp8-atp6-cox3-G-nad3-R-N-A-E-S₁-F-nad5-H-nad4-nad4L-P-T-nad6-cob-S₂-Y-K-Q-C-nad1-L₁-lrRNA-V-srRNA-I-M-nad2-W*).

Model 3, represented by *M. coccopoma*, differs from Model 1 only in the inversion of transposition and a gene cluster reversal (*P-nd4L-nd4-H-nd5-F*). Model 6 and Model 1 differ by one gene cluster (*K-Q-C*) inversion. Model 1, Model 3, and Model 6 have the same TDRL(a) events. Therefore, these three models could have originated from the same ancestor, which is consistent with the phylogenetic tree. Model 2, Model 7, and Model 8 share the same TDRL(b) event, indicating that they could have originated from the same ancestor. Model 4, Model 5, and Model 9 also share the same TDRL(c) event. There are two TDRLs in Verrucomorpha, where only the second differs from TDRL(c). Therefore, we speculated that the second TDRL in Model 10 is associated with the Balanomorpha origin (Figure 4).

*3.6. Phylogeny Analysis*

A phylogenetic reconstruction based on 13 PCGs obtained from the mitochondrial genomes of 44 different Cirripede species is represented in Figure 6. Balanidae, the *M. coccopoma* family, is nested within Pyrgomatidae to form a monophyletic group of Balanoidea (Bootstrap, BP = 100), consistent with previous research [3,45]. The representative of Austrobalanidae (*E. plicata*) is nested within Tetraclitidae, forming a clade sister to Chelonibiidae (*Chelonibia testudinaria*). Together, these two clades develop a monophyletic grouping called Coronuloidea [16,37,46–48]. However, the Catophragmidae belongs to the monophyletic Chthamaloidea and are included with the branches of Balanoidea and Coronuloidea [21]. More data can shed light on the phylogeny of Balanomorpha and the families of this superorder, such as the Chthamalidae and Catophragmidae [47–50]. Chionelasmatidae (*E. ohtai* and *E. coreana*) forms a monophyletic group. [20,51] The Pollicipedomorpha species observed in the intertidal zone (*Capitulum mitella* and *Pollicipes polymerus*) develop a monophyletic group [52,53]. Four species of Lepadomorpha (*Glyptelasma annandalei* (Poecilasmatidae), *Lepas australis* (Lepadidae), *Lepas anatifera* (Lepadidae), and *Lepas anserifera* (Lepadidae) create a monophyletic group at the bottom of the phylogenetic tree with high bootstrap values (BP = 100). The two deep-sea Scallomorpha (*Vulcanolepas fijiensis* and *Arcoscalpellum epeeum*) created separate branches and did not cluster together, inconsistent with previous findings [54,55]. The mitochondrial genome of Balanomorpha is derived from 34 species. In contrast, Verrucomorpha, Scalpellomorpha, and Pollicipedomorpha have only nine species [56]. Therefore, the evolution of Verrucomorpha Verrucomorpha, Scalpellomorpha, and Pollicipedomorpha must be further verified.

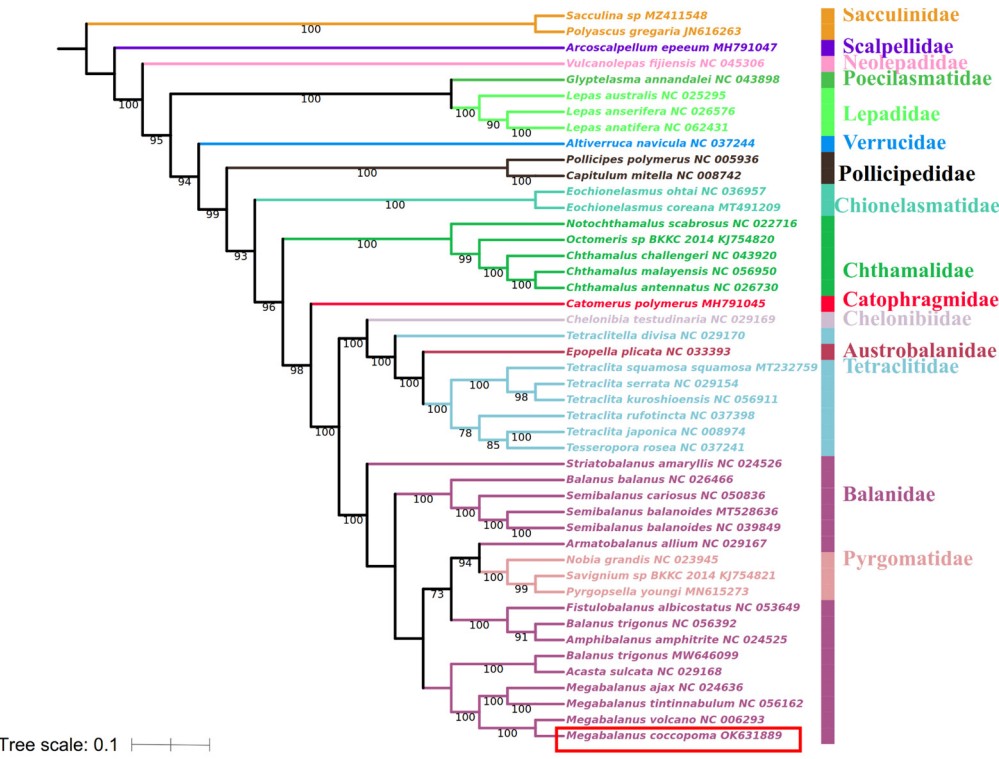

**Figure 6.** The IQ-Tree is based on 13 PCGs nucleotide sequences of *M. coccopoma* and other mitochondrial genomes from 44 Cirripede species. The numbers at the nodes represent the bootstrap values obtained from ML analysis. The IQ-TREE does not take branch length into account. Species in this study are in the red box.

## 4. Discussion

The barnacle mitogenome ranges from 14,906 bp (*Chelonibia testudinaria*) to 17,374 bp (*V. fijiensis*) [54]. The mitochondrial genome length of *M. coccopoma* (15,098 bp) is within this range. The mitogenome of *M. coccopoma* contained 37 genes typical of the metazoan mitogenome. *lrRNA* and *srRNA* are situated in the light strand of the mitochondrial genome and are separated by *V*. This rRNA arrangement is consistent with other Balanomorpha species, except for *Chthamalus challengeri*, *C. malayensis*, and *Chthamalus antennatus*. The rRNAs of these three Chthamalidae species are on the heavy strand, where *srRNA* and *lrRNA* are separated by *V* [17]. Similarly, in the stop codon for *M. coccopoma*, each protein-coding gene is the complete terminator TAA, TAG, or terminated by the incomplete terminator T-. In contrast, the start codon preference uses ATN (N=A, T, C, G); notably, *cox1* starts with CGA, and *nd4L* starts with GTG. The same is true for *Tetraclitella divisa*, *E. plicata*, *M. ajax*, *M. tintinnabulum*, and *M. volcano* [43]. The total number of *M. coccopoma* codons utilized was 3663, with AUU, UUU, and UUA being the most frequently used. CGC was the least frequently used codon in *M. coccopoma*. The most commonly involved amino acids were leucine (Leu), serine (Ser), phenylalanine (Phe), and isoleucine (Ile), as in most Balanomorpha species. The *M. coccopoma* mitochondrial whole genome was rich in AT, with the highest AT content in *lrRNA* (73.0%) and the least in *cox3* (63.4%). This could be due to the depletion of synthetic nucleotides. Nucleotides G and C synthesis requires adequate resources, and this selection leads to efficient resource use for an adaptive evolutionary choice [57].

The gene arrangement of *M. coccopoma* is similar to *Megabalanus* mitogenomes with four translocations (*A*, *S₁*, *T*, *Q*, and *C*), two inversions (*Y* and *K*), and seven conserved gene blocks (*nad6-cob-S₂*, *nad1-L₁-lrRNA-V-srRNA-I*, and *M-nad2-W*), [42]. The conserved blocks between different families were determined for comparison with the original gene arrangement of crustaceans due to the specificity of the mitochondrial gene ring structure.

Still, the evolutionary tree results did not cluster based on these blocks. In our mitogenome phylogenetic tree, *M. coccopoma* clusters in a single clade with other *Megabalanus* species. Balanidae and Pyrgomatidae are nested within each other, creating the Balanoidea clade. The species of Balanoidea experienced the same TDRL(a) event, and the phylogenetic relationships can be associated with gene rearrangement models. Thus, it could be assumed that the same mechanism occurred when the mitochondrial genome was rearranged from the original arrangement among the same families. Our results supported the recent taxonomic revision incorporating the "Archaeobalaninae" subfamily into the Balanidae after undergoing the same rearrangement [1]. The rearrangement mechanism in the Coronuloidea and Chthamaloidea also presents the same way between species families within the two superfamilies. The Coronuloidea species experienced the same TDRL(b) event, and the Chthamaloidea species experienced the same TDRL(c) event (Figure 4). Therefore, the phylogenetic tree based on the mitogenome revealed that Catophragmidae species (*C. polymerus*) formed an independent branch and was situated basal to Pyrgomatidae, Balanidae, Tetraclitidae, and Austrobalanidae. The results were inconsistent with the previous pattern addressed from the phylogenetic analysis of multiple markers [50]. The rearrangement of *C. polymerus, Eochionelasmus coreana, Eochionelasmus ohtai,* and *Octomeris sp.* in Model 4 (Figure 3) experienced the same TDRL(c) event as other Chthamalidae species, including *Notochthamalus scabrosus* (Model 9), *C. antennatus* (Model 5), *C. challengeri* (Model 5), and *C. malayensis* (Model 5) [51,58]. Mitochondrial genome rearrangement indicates that Catophragmidae belongs to Chthamaloidea, consistent with previous findings [59–61]. However, species data are limited for these two superfamilies, and validation requires additional data.

A comparison of Balanomorpha mitochondrial gene rearrangements indicated that the gene arrangement order is relatively conserved within subfamilies but not at the superfamily level. The mitochondrial genome rearrangement could better represent the relationship between Balanomorpha families, particularly between the superfamilies. We discovered one conserved gene block in the balanomorphan species (*M-nad2-W-cox1-L$_2$-cox2-D-atp8-atp6-cox3-G-nad3-R-N-A-E-S$_1$*). Future studies should investigate the primitive arrangement of the Balanomorpha mitochondrial genome based on this conserved gene block to determine the gene rearrangement pattern and the evolutionary relationships with other suborders.

In the phylogenetic tree, Pollicipedomorpha and Verrucomorpha created two separate branches. The two deep-sea scalpellomorphan *Vulcanolepas fijiensis* and *Arcoscalpellum epeeum* clustered into different branch groups, contrary to earlier studies [54]. We attempted to analyze the arrangement between these species and examine whether each species had a different arrangement, more gene rearrangements, gene deletions, and duplications than in the Balanomorpha. Therefore, further species data are required for insights into the Thoracicalcarea phylogeny.

## 5. Conclusions

We described and characterized the full mitogenome of *M. coccopoma*. It contained 37 genes and a putative regulatory area, similar to most metazoan mitogenomes. However, the genes of this species have a completely different order, unlike the crustaceans from which it evolved. We found five gene clusters (or genes) that rearranged based on the pancrustacean ground pattern gene order. These gene clusters contained 10 tRNA genes and 3 PCG genes. The duplication–random loss and recombination theories could explain the large-scale gene rearrangements. The constructed evolutionary trees revealed observations that agreed and disagreed with previous research. Gene cluster (*M-nad2-W-cox1-L$_2$-cox2-D-atp8-atp6-cox3-G-nad3-R-N-A-E-S$_1$*) was arranged in a similar order in the 34 species of Balanomorpha. Therefore, there are basal alignments in the Balanomorpha. This particular linkage may exist between different models, requiring more mitochondrial gene alignments in the balanomorphan species to confirm the pattern.

**Supplementary Materials:** The following supporting information can be downloaded at: https://www.mdpi.com/article/10.3390/d15010117/s1, Table S1: Mitogenome sequences obtained from GenBank for gene arrangement analyses; Table S2: Mitogenome sequences obtained from GenBank for phylogenetic analyses. Table S3: Condon usage in *M. coccopoma*.

**Author Contributions:** Conceptualization, Y.C. and X.S.; methodology, M.Z.; software, M.Z.; validation, M.Z. and N.J.; formal analysis, N.J.; investigation, Y.C.; resources, M.Z.; data curation, M.Z.; writing—original draft preparation, M.Z.; writing—review and editing, Y.C. and B.K.K.C.; visualization, M.Z. and N.J.; supervision, X.S.; project administration, Y.C.; funding acquisition, X.S. All authors have read and agreed to the published version of the manuscript.

**Funding:** This research was funded by the National Natural Science Foundation of China, grant number 41876147, and the Outstanding Youth Foundation of Jiangsu Province, grant number BK20190048.

**Institutional Review Board Statement:** Not applicable.

**Informed Consent Statement:** Not applicable.

**Data Availability Statement:** Mitogenome sequences of *Megabalanus coccopoma* are available in GenBank with accession numbers: OK631889.

**Acknowledgments:** The authors acknowledge any support given which is not covered by the author contribution or funding sections.

**Conflicts of Interest:** The authors declare no conflict of interest. The funders had no role in the design of the study; in the collection, analyses, or interpretation of data; in the writing of the manuscript; or in the decision to publish the results.

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
