# Peer review of "The Mitochondrial Genome of the Globally Invasive Barnacle Megabalanus coccopoma Darwin 1854 (Crustacea: Balanomorpha): Rearrangement and Phylogenetic Consideration within Balanomorpha"

_diversity, doi:10.3390/d15010117_

Round 1
Reviewer 1 Report
I have read the MS with curiosity and interest and found some hypotheses very appealing. Besides the exhaustive analysis of the mitochondrial genome of a cosmopolitan (invasive) species of barnacle, results also have phylogenetic implications for the vast taxon Cirripedia. Poor English (grammar and verbosity) plagues the text so much that I may have missed some crucial points. Consequently, in all faith, I recommend this article for publication after a rigorous correction of the English text.
Along is a pdf file with embedded some suggested corrections.

Author Response
Response to Reviewer 1
Question 1: Line 38:delete “due to the diversity of its habitat, and”
Response: We have deleted “due to the diversity of its habitat, and”
Question 2: Line 40 species and the upper-order elements of species “taxta?”
Response: We have rewritten “taxonomy” as “taxta”
Question 3:Line 43 “In addition,” rewrite as“However”
Response: We have rewritten “In addition” as “However”
Question 4:Line 46 “There are 37,” rewrite as“In general, there are...”
Response: We have rewritten “There are ” as “In general, there are”
Question 5:Line 52 “has also changed.” rewrite as“may also vary.”
Response: We have rewritten “has also changed.” as “may also vary.”
Question 6:Line 59 “compared with Pollicipes” rewrite as“compared it with those of ”
Response: We have rewritten “compared with Pollicipes” as“compared it with those of ”
Question 7: Line 65 original “plesiomorphic (?)”
Response: We have rewritten “original” as “plesiomorphic ”
Question 8: Line 69-70 “serrata. They based on gene alignments and phylogenetic analyses to demonstrate that the Bal-” rewrite as “based on ..... demostrated the non-monophyletic nature of Balanidae”
Response: We have rewritten “They based on gene alignments and phylogenetic analyses to demonstrate that the Balanidae family is not monophyletic.” as“and based on gene alignments and phylogenetic analyses to demonstrate the non-monophyletic nature of Balanidae.”
Question 9: Line 71 “Due” rewritten as“Yet, due to”
Response: We have rewritten “Due” as “Yet, due to ”
Question 10: Line 71 “the evo-” rewritten as“their evolutionary tree, further studies are needed to confirm the ....”
Response: We have rewritten “the evolutionary tree, it is necessary to confirm t” as “the low self-extension values within the evolutionary tree, it is necessary to confirm”
Question 11: Lines 75-77 unclear, please rephrase the sentence in accordance with the previous one.
Line 78-79 unclear, please rephrase the sentence in accordance with the previous one.
Response: We have rephrased.“The pattern of mitochondrial gene arrangement in Balanomorpha remains unknown. This is supported by the growing body of evidence pointing to an ancestral arrangement of mitochondrial genes in the Balanomorpha. [17]. In the meantime, Mitochondrial gene arrangement has been shown to address phylogenetic studies effectively [18]. Therefore, the Balanomorpha mitochondrial genome rearrangement study will provide vital information on the profound systemic evolution within the Balanomorpha collective group [19-21]. However, the data on Balanomorpha mitochondrial genome is still scarce, which seriously restricts the deep discussion of the systematic evolution of various groups [22, 23].”
Question 12: Line 83-87 Please consider. In this study we i) sequenced the entire mitogenome of...
- ii) addressed the gene rearrangement in ... and iii) otained a Toracicalcarea
Response 12:We have rephrased. “In this study, we i) sequenced the entire mitogenome of an M. coccopoma individual and provided a comprehensive description of our findings, ii) addressed the gene rearrangement in Balanomorpha's mitogenome and iii) obtained a Thoracicalcarea phylogenetic tree to determine M. coccopoma's relationship to other barnacles.”
Question 13: Line 72 “currently” rewritten as“currently it is kept”
Response: We have rewritten “currently” as “currently it is kept ”
Question 14: Line 98-99 “general” rewrite as“universal (?)”, “particular” rewrite as“specific (?)”, “to strengthen” rewrite as“amplify (?)”
Response: We have rewritten “general” as “universal”, “particular” as “specific”, and “strengthen” as “amplify”.
Question 15: Line 109-110 Please move this part up, should be placed soon after the collection
Response: We have moved this paragraph to line 95.
Question 16: Line 143 “primitive” rewrite as“plesiomorphic”
Response: We have rewritten “primitive” as “plesiomorphic ”
Question 17: Line 144 “The results of the CREx [31] investigation 144 were gene rearrangements.” unclear to me!
Response: We have rewritten it as “Gene rearrangements were analyzed by CREx [31] for output. ”
CREx is a web-based application for analyzing gene orders based on the application server Zope. It can output the results of the analysis of mitochondrial genome rearrangements. After uploading the gene order data in FASTA format to CREx, a matrix is computed and displayed, with elements colored according to an evolutionary distance measure, such that gene orders with a small evolutionary distance can be easily identified. Therefore, the breakpoint distance, the reversal distance or the number of common intervals is utilized.
Question 18: Line 158-159 “ Of the 13 PCGs genes, 22 tRNA genes and 2 rRNA genes exist. The numbers” rewrite as“Beside 13 PCGs there are 22 tRNA and 2 rRNA genes. ; complete details are reported in...”
Response: We have rewritten “Of the 13 PCGs genes, 22 tRNA genes and 2 rRNA genes exist. The numbers” as “Besides 13 PCGs genes, 22 tRNA genes and 2 rRNA genes exist; complete details are reported in (Table 2, Fig 1). ”
Question 19: Line 161 “fragile” rewritten as“Light”
Response: We have rewritten “fragile” as “Light ”
Question 20: Line 174 “M. coccopoma”No italics
Response: We have changed it to italic and modified Fig.1.
Question 21: Line 180-181 but this has been said previously already (see above)
Response: We have removed this sentence
Question 20: Line 182-192 Unclear, please rephrase these sentences.
Response: We have rewritten it as “All 13 PCGs in M. coccopoma start with ATN (N=A, T, G, and C), except for nd4L, which has GTG as its start codon, which reflects the conservation of genes in Balanomorpha genes. In addition, five PCGs (cox3, nd3, nd5, nd4, and cob) terminate with incomplete terminators (TA- or T-), while the remaining PCGs are terminated with complete terminators TAA termination. (Table 2). Incomplete terminators are prevalent in Balanomorpha [15, 38, 39]. The 13 PCGs of M. coccopoma have 3,663 codons (not counting the incomplete terminators), with the 6 most frequent ones being UUU> AUU> UUA> AUA> UCU> UUC, was 2.94% to 7.05% and the lowest being CGC (0.11%). The multiple codons are UUU (259), followed by AUU (257) (Fig. 2, Supplementary Table S3). The most commonly used amino acids were leucine (Leu) (544), phenylalanine (Phe) (367), serine (Ser) (359), and isoleucine (Ile) (317).”
Question 21: Line 193frequency (?), and what is the label for the y axis?
Response: We have rewritten as “Codon usage in the 13 mitochondrial protein-coding genes of M. coccopoma” Number of codons is the label for the y axis.
Question 22: Line 207 “to studies,” rewrite as“Previous studies indicated that...”
Response: We have rewritten “According to studies” as “Previous studies indicated that ”
Question 23: Line 212 “clover ” rewritten as“clover leaf”
Response: We have rewritten “clover” as “clover leaf ”
Question 24: Line 214 at 748bp I am not sure it is correct to position a gene in term of bp.
Response:We are particularly sorry, this is a mistake in our presentation, this should be the length of the gene. We have changed it to "748 bp long" and "1,319 bp long"
Question 25: Line 38:delete “For the phylogenetic study, particularly those focusing on links between ancestral 223 arrangement order, a comparison of gene permutations may be helpful [21]. Crustaceans 224 and hexapods display the traditional mitochondrial gene arrangement [42].”
Response: We have deleted “For the phylogenetic study, particularly those focusing on links between ancestral 223 arrangement order, a comparison of gene permutations may be helpful [21]. Crustaceans 224 and hexapods display the traditional mitochondrial gene arrangement [42].”. And the references have been amended.
Question 26: Line 65 original “plesiomorphic (?)”
Response: We have rewritten “original” as “plesiomorphic ”
Question 27: Line 227- 228 erase and replace with, yielding ten different types, each with....
Response: We have rewritten “Based on the mitochondrial genome sequence of 35 barnacle species and classified into ten types, each with its unique conserved gene block.” as “, yielding ten different types, each with its unique conserved gene block.”
Question 28: Line 234 all of the above?please put a dot . after RNAs and convoy your meaning in a new sentence.
Line 236 240-242 Unclear! please state clearly if a give type is conserved at the genus or family level. An provide examples in support. Please notice that you may say that a give type is conserved only if you have analysed two or more
Line 244 please state what is the consistence you are referring to.
Response: We have rewritten this paragraph to show that the types of Model 1 and Mode 4 are not conservative at the level of the family. The consistence you are referring to the phylogenetic tree, the different gene arrangements are also well clustered, Balanomorpha genes is consistent with the results shown by the phylogenetic tree.
“This gene rearrangement is in agreement with previous studies [42-44]. The Balanidae, in which M. coccopoma is found, contain three Models: Model 1, Model 3 and Model 6. M. coccopoma shares Model 3 with Balanidae (M. volcano, Megabalanus tintinnabulum, Megabalanus ajax, Acasta cyathus, Acasta sulcata and Balanus trigonus). The Pyrgomatidae are Model 1. The Tetraclitidae are predominantly Model 2 in this gene arrangement, Austrobalanidae (Model 7), Chelonibiidae (Model 8), Catophragmidae (Model 4), Chthamalidae (Model 4, Model 5 and Model 9), Chionelasmatidae(Model 4), Verrucidae(Model 10). In addition, the ten-types gene arrangement order is only conserved in the genera, not conservative between different families, for example, the Balanidae (Armatobalanus allium, Striatobalanus amaryllis, Amphibalanus amphitrite, Balanus Balanus, Fistulobalanus albicostatus and Semibalanus cariosus) and the Pyrgomatidae (Nobia grandis, Pyrgopsella youngi, and Savignium sp. BKKC-2014) show the same gene arrangement model (Fig. 3, Model 1); the Chthamalidae (Octomeris sp. BKKC-2014), the Catophragmidae (Catomerus polymerus) and the Chionelasmatidae (Eochionelasmus ohtai and Eochionelasmus coreana) show the same gene arrangement model (Model 4). It is interesting to note that in the phylogenetic tree, the different gene arrangements are also well clustered, for example, Balanus trigonus (Model 3)与 A. amphitrite (Model 1) cluster together; Semibalanus cariosus (Model 1) and Striatobalanus amaryllis (Model 1) cluster together, indicating that the order of Balanomorpha genes is consistent with the results shown by the phylogenetic tree (Fig. 4).”
Question 29: Line 268-269 we found that the order also, it is not clear top me what trapezoids are
Response: We have rewritten “this paper finds .as “we found”.We are really sorry, the word should be Balanomorpha. We have rewritten “trapezoids” as “Balanomorpha ”
Question 30: Line 212 “cluster” rewrite as“cluster the taxa according to...”
Response: We have rewritten “cluster ”as “cluster the taxa according to ”
Question 31: Line 272-273 Unclear. Line 280 “imply” rewrite as “determine (?)”
Response: We have moved this paragraph to line 280 for explanation, where "imply" rather than "determine" would better convey the meaning of the sentence.
“In many cases, repeated random loss (TDRL) events could be implied the evolutionary direction of rearrangements and allow the reconstruction of ancestral states by comparing two gene orders without accounting for outgroups.”
Question 32: Line 275:delete “frequently used”
Response: We have deleted “frequently used”
Question 33: Line 277 “method” rewritten as“mechanism (?)”
Response: We have rewritten “method” ”as “mechanism”
Question 34: Line 278 Line 278 “Duplication” Consequently, duplication-random.....”
Response: We have rewritten “Duplication” ”as “Consequently, duplication-random”
Question 35: Line 282 “this study” rewritten as“we investigated”
Response: We have rewritten “this study” ”as “we investigated”
Question 35: Line 283 “gene” rewritten as“gene, focussing on TDRL event, in ”
Response: We have rewritten “gene” ”as “gene, focussing on TDRL event, in ”
Question 36: Line 284-286:delete “We are 284 concentrating on the TDRL events in the mitochondrial genomes of 34 different species of 285 Balanomorpha.”
Response: We have deleted “We are 284 concentrating on the TDRL events in the mitochondrial genomes of 34 different species of 285 Balanomorpha.”
Question 37: Line 292 “original” rewrite as“plesiomorphic ”, delete “primitive”
Response: We have rewritten “original” as “plesiomorphic ” and deleted “primitive”.
Question 38: Line310 make a different paragraph
Response: We have started another section
Question 39: Line310 “where” rewrite as“shown by ”,“is located and” rewrite as“diffe by”
Response: We have rewritten it as “Model 3 shown by M. coccopoma differs by Model 1 differs only in the inversion of transposition and reversal of a gene cluster (P-nd4L-nd4-H-nd5-F).”.
Question 40: Line321-322 “model” rewrite as“phylogenetic reconstruction based on ”, “was shown” rewrite as“is shown”
Response: We have rewritten “model” as“phylogenetic reconstruction based on ”, “was shown” as“is shown”
Question 41: Line324-326 the representative of Austrobalanidae... is nested within Tetraclitidae, making up a clade sister to Chelonibiidae (....); all together these two clades form a monophyletic grouping called Coronuloidea (?).
Response: We have rewritten it as “The representative of Austrobalanidae (Epopella plicata) is nested within Tetraclitidae, making up a clade sister to Chelonibiidae (Chelonibia testudinaria); altogether these two clades form a monophyletic grouping called Coronuloidea [16, 38, 47-49]”.In the meantime we have added a description.“Chionelasmatidae (Eochionelasmus ohtai and Eochionelasmus coreana) form a monophyletic group.”
Question 42: Line 327 329:delete “Tetraclitidae, Austrobalanidae and Chelonibiidae,”, “More data and research are needed to learn more” rewrite as “shed light on”
Response: We have rewritten it as “More data are needed to shed light on the phylogeny of Balanomorpha and the families that make up this superorder, including the Chthamalidae and Catophragmidae”
Question 43: Line 331 332 333 335 “in” rewrite as “found in the intertidal zone”, “formed” rewrite as “form”, “form” rewrite as “of”, “formed” rewrite as “form”.
Response: We have rewritten “in,” as “found in the intertidal zone”, “formed” as “form”, “form” rewrite as “of”, and “formed” as “form”.
Question 44: Line 338-340:Unclear! Do you mean that the mitochondrial genome is know for 34 species of Balanomorpha, nine species of Verrucomorpha etc...?
Response: We have rewritten this paragraph to be Verrucomorpha, Scalpellomorpha and Pollicipedomorpha have only nine species.
“The mitochondrial genome of Balanomorpha has 34 species; in contrast, Verrucomorpha, Scalpellomorpha and Pollicipedomorpha have only nine species [57]. The evolution of more Verrucomorpha Verrucomorpha, Scalpellomorpha and Pollicipedomorpha needs to be further verified.”
Question 45: Line 349-350 Add“The mitochondrial genome length”, “mitogenomes” rewrite as “mitogenome”
Response: We have changed the sentence to“The mitochondrial genome length of M. coccopoma (15,098 bp) is located within this range. The mitogenome of M. coccopoma contained 37 genes which are typical of the metazoan mitogenome, where lrRNA and srRNA are located in the light strand of the mitochondrial genome, separated by V.”
Question 46: Line 353-354 “except for Chthamalus challengeri, C. malayensis, and Chthamalus 353 antennatus” please state in what consists the exception
Response: We have stated that this exception includes the rRNAs of these three Chthamalidae species on the heavy strand, where srRNA and lrRNA separated by V
Question 47: line 354 “with” rewrite as “to”
Response: We have rewritten “with” rewrite as “to”
Question 47: line 481 this seems a wrong doi, please check it out. https://doi.org/10.1146/annurev.es.18.110187.001413
Response: We have changed to the correct doi and corrected the errors that occurred in other references

Reviewer 2 Report
This manuscript investigated the entire mitogenome of an Megabalanus coccopoma individual and provided a comprehensive description. A large-scale gene rearrangement in Balanomorpha's mitogenome was also studied and the Thoracicalcarea phylogenetic tree was reconstructed systematically to determine M. coccopoma's relationship to other barnacles. As their described the results showed that the order of gene arrangement in trapezoids is highly differentiated, which has been hypothesized that this could be because specific evolutionary paths could be implied between the gene rearrangements of the Balanomorpha. As well, Balanomorpha can be divided into 3 TDRL events, as TDRL(a), TDRL(b) and TDRL(c), while Verrucomorpha is TDRL(d) through concentrating on the TDRL events in the mitochondrial genomes of 34 different species of Balanomorpha. Overall, the topic of the paper is of interest for the scientific community and the paper is well-written and clear. The manuscript has met the requirements for publication. However, the paper needs to be improved before acceptance for publication, as indicated below.
Minor comments
Question 1
Language errors throughout this manuscript should be revised, such as structure, improper use of capital letters, abbreviations and italic formatting.
Question 2
There are some errors in the format of the references section. Please modify it. Such as 9 and 10...
Question 3
We noticed that the GenBank ID (OK631889) mentioned in the text is not active. Please ensure that all accession numbers associated with your submission are activated.
Author Response
Response to Reviewer 2
Question 1
Language errors throughout this manuscript should be revised, such as structure, improper use of capital letters, abbreviations and italic formatting.
Response: We apologize for the language problems in the original manuscript. The language presentation was improved with assistance from a native English speaker with appropriate research background.
Question 2
There are some errors in the format of the references section. Please modify it. Such as 9 and 10...
Response: We apologize for the errors format of the references section. We have revised the formatting of those incorrect references.
Question 3
We noticed that the GenBank ID (OK631889) mentioned in the text is not active. Please ensure that all accession numbers associated with your submission are activated.
Response: We are grateful for the suggestion. We have ensured that all accession numbers associated with our submission are activated.

Reviewer 3 Report
An interesting contribution to the knowledge of the genom of a widely distributed balanomorph, Just marked a few points to be corrected.

Author Response
Response to Reviewer 3
Question 1: Balanomorph Initial letters should be lowercase
Response: We've lower-cased the Balanomorph initials and checked for other occurrences of the error.
Question 2: specie should be in the plural
Response: We have changed the name of the species from "specie" to “Altiverruca navicular”.
Question 3: Line 174 “M. coccopoma”No italics
Response: We have changed it to italic and modified Fig.1.
Question 4: Line 352 Chthamalus antennatus written as an abbreviation for C. antennatus
Response: This is the first time this species has appeared it should be given its full name. We have checked elsewhere where abbreviations are needed and made changes.
Question 5: Line 358 “and” not italics
Response: We have changed it to un-italicised and checked for italic issues elsewhere.
Question 6: Line 438 delete extra dot
Response: We have removed the redundant points and checked the rest.
